# Highly Sensitive and Specific Detection of Mobilized Colistin Resistance Gene *mcr-1* by CRISPR-Based Platform

ⓘLin Gong,[a] Zhengjiang Jin,[b] Ernan Liu,[a] Fei Tang,[c] Fengyun Yuan,[c] Jiansheng Liang,[a] Yimei Wang,[a] Xiaoli Liu,[a] Yi Wang[d]

[a]Department of Disinfection and Pest Control, Wuhan Center for Disease Control and Prevention, Wuhan, Hubei, China
[b]Department of Clinical Laboratory, Maternal and Child Health Hospital of Hubei Province, Wuhan, Hubei, China
[c]Institute of Environmental Medicine, MOE Key Laboratory of Environment and Health, School of Public Health, Tongji Medical College, Huazhong University of Science and Technology, Wuhan, Hubei, China
[d]Experimental Research Center, Capital Institute of Pediatrics, Beijing, China

Lin Gong, Zhengjiang Jin and Ernan Liu contributed equally to this work. Author order was determined on the basis of seniority.

**ABSTRACT** Mobilized colistin resistance (*mcr-1*) gene mediated by plasmid can cause the speediness dissemination of colistin-resistant strains, which have given rise to a great threat to the treatment of human infection. Hence, a rapid and accurate diagnosis technology for detecting *mcr-1* is essential for the control of resistance gene. Here, a recombinase polymerase amplification (RPA) coupled with CRISPR/Cas12a platform was established for rapid, sensitive, and specific detection of *mcr-1* gene. The analytical sensitivity of our assay is 420 fg per reaction in pure *mcr-1*-positive isolates, and the threshold of this method in spiked clinical samples was down to $1.6 \times 10^3 \sim 6.2 \times 10^3$ CFU/mL ($1.6 \sim 6.2$ CFU/reaction). Moreover, the RPA-CRISPR/Cas12a system perspicuously demonstrated no cross-reactivity with other resistant genes. The entire experimental process included rapid DNA extraction (15 min), RPA reaction (30 min), CRISPR/Cas12a cleavage (5 min), and fluorescence testing (<10 min), which could be completed within 60 min. In summary, the RPA-CRISPR/Cas12a assay designed here provides a rapid diagnostic way for monitoring *mcr-1* in clinic and livestock farm.

**IMPORTANCE** This study promises a rapid and accurate assay (RPA-CRISPR/Cas12a) for the surveillance of *mcr-1* gene, which causes the efficacy loss of colistin in clinical treatments. In addition, the established method is fit for "on-site" surveillance especially.

**KEYWORDS** *mcr-1*, colistin resistance, RPA, CRISPR, Cas12a

In recent decades, a huge surge in serious infections caused by carbapenem-resistant *Enterobacteriaceae* (CRE) gives rise to severely threat to human life (1). Because of contraindication and ineffectiveness of other antibiotics, colistin is regarded as a last resort for treatments of clinical infections (2). However, the augmenting consumptions of colistin have led to the emerging risk of drug resistance.

Strains utilize several resistance mechanisms to resist colistin, including modified efflux pump and lipopolysaccharide (3). In 2015, a new resistance way involving lipopolysaccharide modification named mobilized colistin resistance (*mcr*) was first reported in China, and the *mcr-1* gene was initially detected in *Escherichia coli* from humans and animals (4). Afterwards, this resistant gene has been observed around the world across six continents, such as Colombia, Japan, America, Brazil, Denmark, New Caledonia, and Egypt (5, 6). Meanwhile, the *mcr-1* has been found in all kinds of bacterial species, including *Escherichia fergusonii*, *Citrobacter braakii*, *Klebsiella pneumoniae*, *Salmonella enterica*, *Klebsiella oxytoca*, *Acinetobacter baumannii* and *Kluyvera ascorbata* (6). Benefiting from the high transferability, the *mcr-1* gene is widely identified in humans (healthy human and clinical patients) (7,

Address correspondence to Xiaoli Liu, liuxiaoli20851@126.com, or Yi Wang, wildwolf0101@163.com.

The authors declare no conflict of interest.

8), animals (food animals, pets, and wild animals) (9–11), and environmental settings (feces, manure, farm soil, rivers, and oceans) (6). Comparing with natural water, farm, and urban environments have higher prevalence rates of *mcr-1*-carrying strains (6). The horizontal transmission of *mcr-1* gene can be accomplished by various types of plasmids including *IncP1*, *IncX4*, *IncN*, *IncF*, *IncFIB*, *IncI2*, *IncFII*, and *IncHI2* (12, 13). Therein, the *IncX4*, *IncI2*, and *IncHI2* plasmids are dominant types (13, 14). In addition to horizontal spread, vertical propagation also plays an important role in *mcr-1*-positive isolates (6). The same sequence typing (ST) and pulsed-field gel electrophoresis (PFGE) patterns of *mcr-1*-producing strains detected in many studies support this viewpoint (15, 16). Global dissemination of *mcr-1* might cause the means shortage for the treatments of CRE isolates infections. Meanwhile, several methods, mostly PCR, have been developed for *mcr-1* diagnosis (4). However, these technologies are time-consuming and insensitivity. Therefore, a rapid and accurate technology for *mcr-1* detection contributes to reducing the overuse of antibiotics.

The CRISPR/CRISPR-associated protein (Cas) as a system was originally found and whereafter was taken as an useful tool for genome editing and nucleic acid diagnosis owing to its reliability, sensitivity, and specificity (17). Multiple types of CRISPR/Cas systems including Cas9, Cas12a, Cas12b, Cas13 and Cas14 are successively developed for bacteria, viruses, and parasites detections (17, 18). Especially in CRISPR/Cas12a system, the guide RNA (gRNA) firstly guides Cas12a enzyme to recognize the target nucleic acid. After the gRNA binds the matching sequence nearing protospacer adjacent motif (PAM) site (TTT), the enzyme shears the target DNA. Simultaneously, nonspecific and collateral capacity of the Cas12a enzyme are triggered to cleave surrounding ssDNA (single-strand DNA) (19). The probe could be made by ssDNA labeled with fluorophore and quencher at both ends, and the degradation of that emits the fluorescent molecule detected by fluorescence detector (20).

To improve the sensitivity of CRISPR/Cas system, the target sequence could be pre-augmented using isothermal amplification technique including loop-mediated isothermal amplification (LAMP), multiple cross displacement amplification (MCDA) and recombinase polymerase amplification (RPA) (21). These approaches yielding amplifcons only need a simple thermostat, omit the sophisticated infrastructures and suit for application in point-of-care detection (22). Combining with isothermal amplification, the Cas12a has been used to design for nucleic acid diagnosis platforms. And the technologies have been applied in the microorganism surveillance such as COVID-19, African swine fever virus (ASFV) and severe fever with thrombocytopenia syndrome virus (SFTSV) (21, 23, 24).

In this study, we coupled RPA with CRISPR/Cas12a system to build a novel method named RPA-CRISPR/Cas12a for the rapid detection of *mcr-1*, the principle and workflow of the assay were illustrated in Fig. 1 and 2, and the specificity and sensitivity of which in axenic culture and simulate samples were elaborated.

## RESULTS

**Schematic plot of *mcr-1* detection based on RPA-CRISPR/Cas12a.** Here, the RPA-CRISPR/Cas12a platform was used for the detection of *mcr-1* gene and the principle of the platform is shown in Fig. 1. In brief, the target gene segment containing a PAM site (TTTC) is amplified by the RPA kit at an equal temperature of 37°C (Fig. 1, steps 1 and 2). Recombinase-primer complex formed by combination of recombinase and specific primer can search for homologous sequence in double-stranded DNA template. When the specific sequence of template is recognized, chain exchange reaction will be initiated, and the target fragment will be amplified exponentially under the action of DNA polymerase. During the detection stage, the gRNA-CRISPR/Cas12a complex recognizes the target sequence by the PAM site (Fig. 1, step 3), which successfully activates the effector molecule Cas12a. Then, the activated Cas12a protein cuts the subject sequences, and simultaneously implements nonspecific trans-cleavage of probe (ssDNA labeled with fluorophores) in reaction system (Fig. 1, steps 4 to 6). Finally, the fluorescent signals are released and can be collected as indicator for the detection results. The whole diagnostic process of RPA-CRISPR/Cas12a assay, including genomic

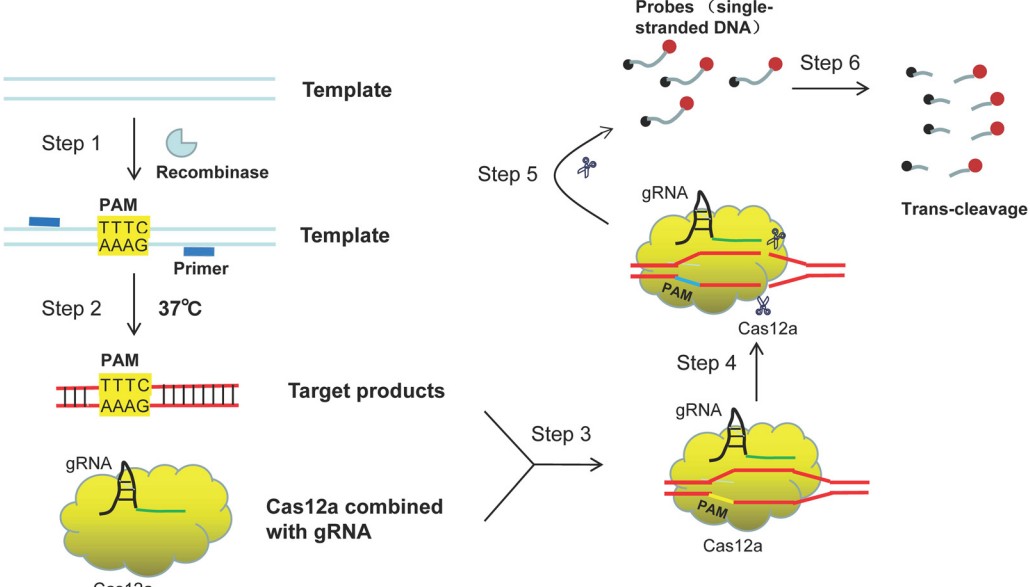

**FIG 1** Outline of the RPA-CRISPR/Cas12a assay for *mcr-1* detection. Target sequence containing a PAM (TTTC) site is specifically amplified by the RPA reaction (steps 1 to 2). When combining with amplicons products, the gRNA-CRISPR/Cas12a complex is induced to nonspecifically cleave the ssDNA (both ends labeled with fluorophores) (steps 3 to 6).

extraction (15 min), RPA reaction (30 min), CRISPR/Cas12a cleavage (5 min), and fluorescence testing (10 min), can be accomplished within 1 h (Fig. 2).

**The verification of *mcr-1* RPA-CRISPR/Cas12a method.** The *mcr-1* RPA were performed at 37°C for 30 min to confirm the reliability of RPA primers. 3 μL of each product was loaded on the 2% agarose gel, and the results of agarose gel electrophoresis showed that positive reaction appeared in the tube loaded with *mcr-1*-positve *E. fergusonii* WH-ZX154, but not with IMP-26-producing *Enterobacter hormaechei* (WHCDC-SP67), qnrS1-carrying *Enterobacter cloacae* (WHCDC-XW38) and distilled water (Fig. S1) .The same results were obtained using fluorescence detection in the following CRISPR/Cas12a assay (Fig. 3). Therefore, the primers of RPA were available for the RPA-CRISPR/Cas12a assay to diagnose *mcr-1* gene.

**Optimal temperature for *mcr-1* RPA-CRISPR/Cas12a assay.** The plasmid DNA of *E. fergusonii* WH-ZX154 (4.2 pg/μL) was used for optimizing the amplification temperature of RPA reaction stage. The temperatures from 35°C to 42°C with 1°C increments were performed to compare the amplifying efficiency of RPA in different conditions. The electrophoresis results manifested that 37°C to 39°C were the better candidates

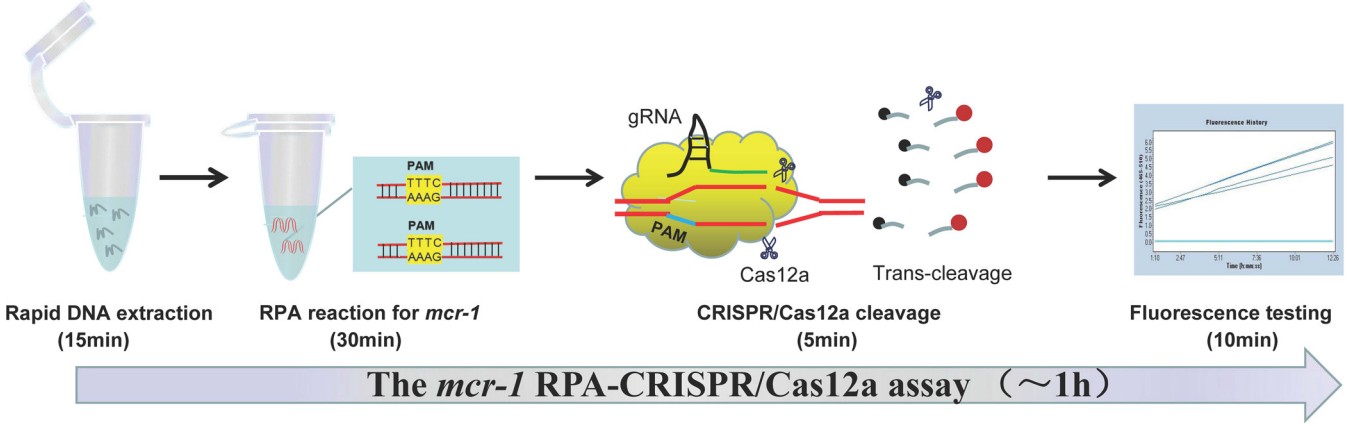

**FIG 2** Schematic of the *mcr-1* RPA-CRISPR/Cas12a assay workflow. The entire workflow of RPA-CRISPR/Cas12a method includes 4 steps: rapid DNA extraction, RPA reaction, CRISPR/Cas12a cleavage, and fluorescence testing.

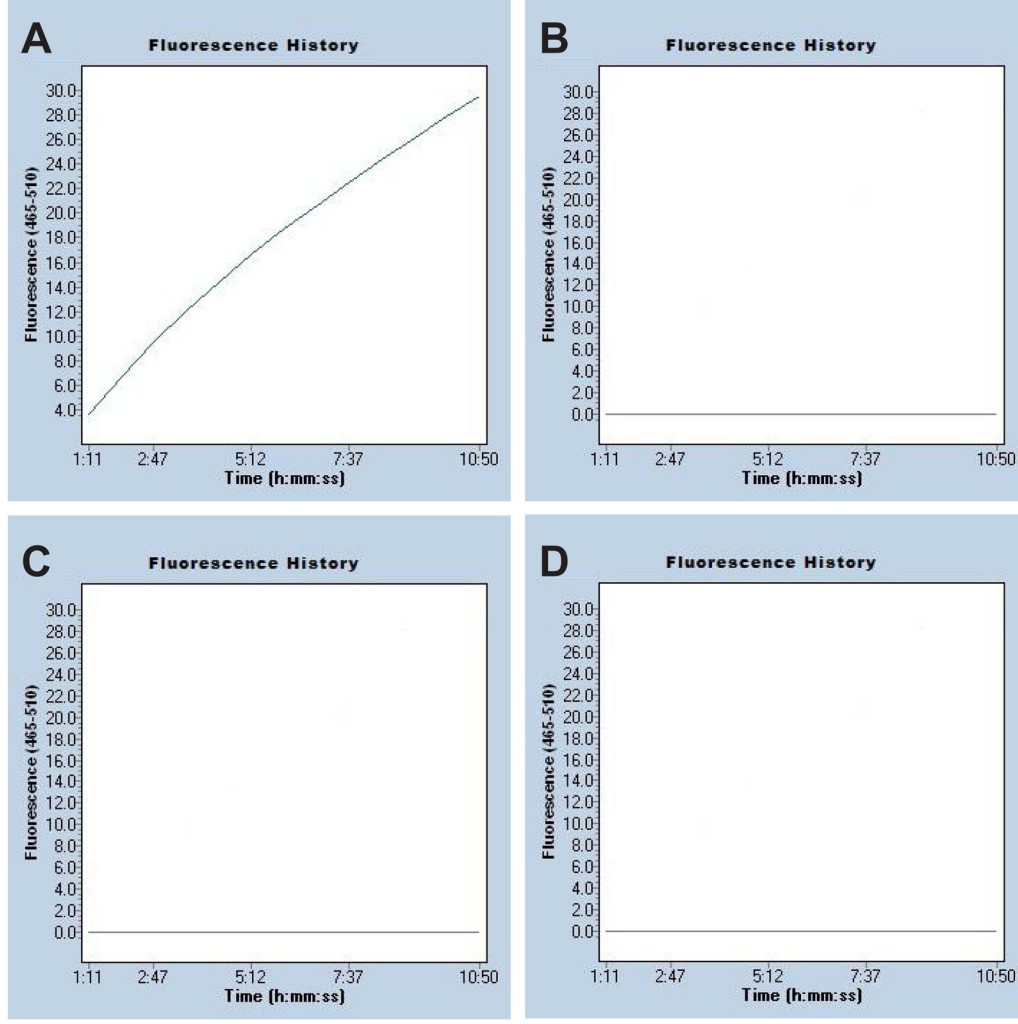

**FIG 3** Confirmation of the *mcr-1* RPA-CRISPR/Cas12a assay. Four representative samples were verified by employing fluorescence detection: (A) *mcr-1*-positive *Escherichia fergusonii* (WH-ZX154); (B) IMP-26-producing *Enterobacter hormaechei* (WHCDC-SP67); (C) qnrS1-producing *Enterobacter cloacae* (WHCDC-XW38); (D) distilled water.

for *mcr-1* RPA reaction (Fig. 4). Hence, 37°C was selected for the subsequent CRISPR/Cas12a tests.

**Sensitivity and specificity estimation of the RPA-CRISPR/Cas12a system for *mcr-1*.** We evaluated the limit of detection (LoD) of the RPA-CRISPR/Cas12a system with serial dilutions of reference strains (*E. fergusonii* WH-ZX154). The CRISPR-*mcr-1* assay was executed as described above, and the results were acquired by detecting fluorescence signals. At last, the system could stably detect even 420 fg of plasmid DNA through three parallel trials (Fig. 5).

To analyze the specificity of the RPA-CRISPR/Cas12a system, DNA from strains carrying a variety of resistant genes (KPC-2, NDM-1, IMP-26, OXA-181, rmtB, qnrS1, TEM-1B and sul1) were extracted. Positive signals were only detected in the *mcr-1*-producing samples compared with these non-*mcr-1* species. The positive predictive value (PPV), negative predictive value (NPV), sensitivity and specificity of the RPA-CRISPR/Cas12a technology were entirely 100% (Table S1). Thus, the developed RPA-CRISPR/Cas12a system could specifically monitor the *mcr-1* gene (Fig. 6).

**The feasibility of RPA-CRISPR/Cas12a system to *mcr-1*-spiked clinical samples.** To examine the practicability of RPA-CRISPR/Cas12a system as a *mcr-1* diagnostic approach, the system was applied in spiked clinical samples added with a series of *mcr-1*-positive strains dilutions. There were 3 duplicates per dilution, and all of $6.2 \times 10^3$ CFU/mL (6.2 CFU/reaction) inputs in stool samples, $2.6 \times 10^3$ CFU/mL (2.6 CFU/reaction) inputs in blood

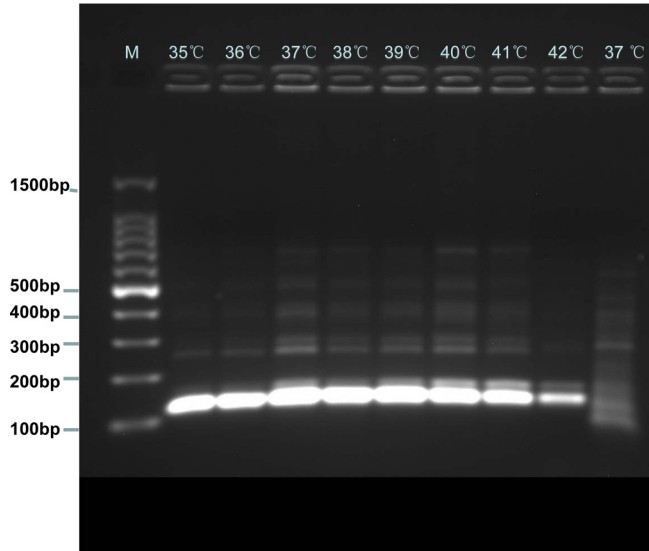

**FIG 4** Optimal temperature for RPA reaction. The RPA products of plasmid DNA of *mcr-1*-positive *Escherichia fergusonii* (WH-ZX154) (4.2 pg/μL) amplified in different temperatures were analyzed using electrophoresis experiment. Lanes 1–8, amplicons electrophoretogram at 35 to 42°C with 1°C increments. Lane 9, blank control.

samples, $1.6 \times 10^3$ CFU/mL (1.6 CFU/reaction) inputs in urine samples were deemed to positive, but the lower concentrations gave inefficient fluorescence signals (Fig. 7 and Fig S2 and S3). The threshold of this method in spiked clinical samples was $1.6 \times 10^3 \sim 6.2 \times 10^3$ CFU/mL (1.6 ~ 6.2 CFU/reaction). Hence, the low-threshold demonstrated the established method was fit for *mcr-1* detection in practice.

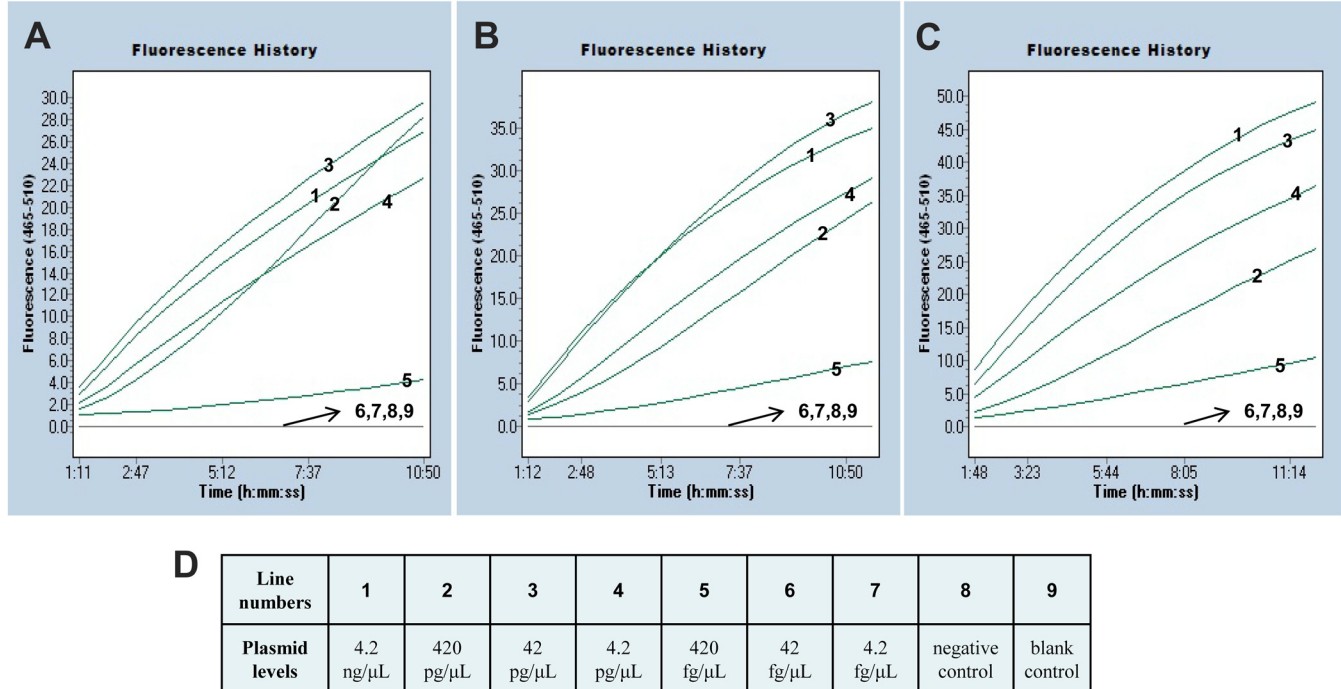

**FIG 5** Sensitivity of the *mcr-1* RPA-CRISPR/Cas12a assay. Figures (A), (B), and (C) displayed the fluorescence detection results of three replicated tests respectively. (D) Showed the line number matching each plasmid concentration, negative and blank control. Lines 1 to 9 denoted the plasmid levels (*Escherichia fergusonii* WH-ZX154) of 4.2 ng, 420 pg, 42 pg, 4.2 pg, 420fg, 42fg, 4.2fg per reaction, negative control (IMP-26-producing *Enterobacter hormaechei* WHCDC-SP67), and blank control, respectively.

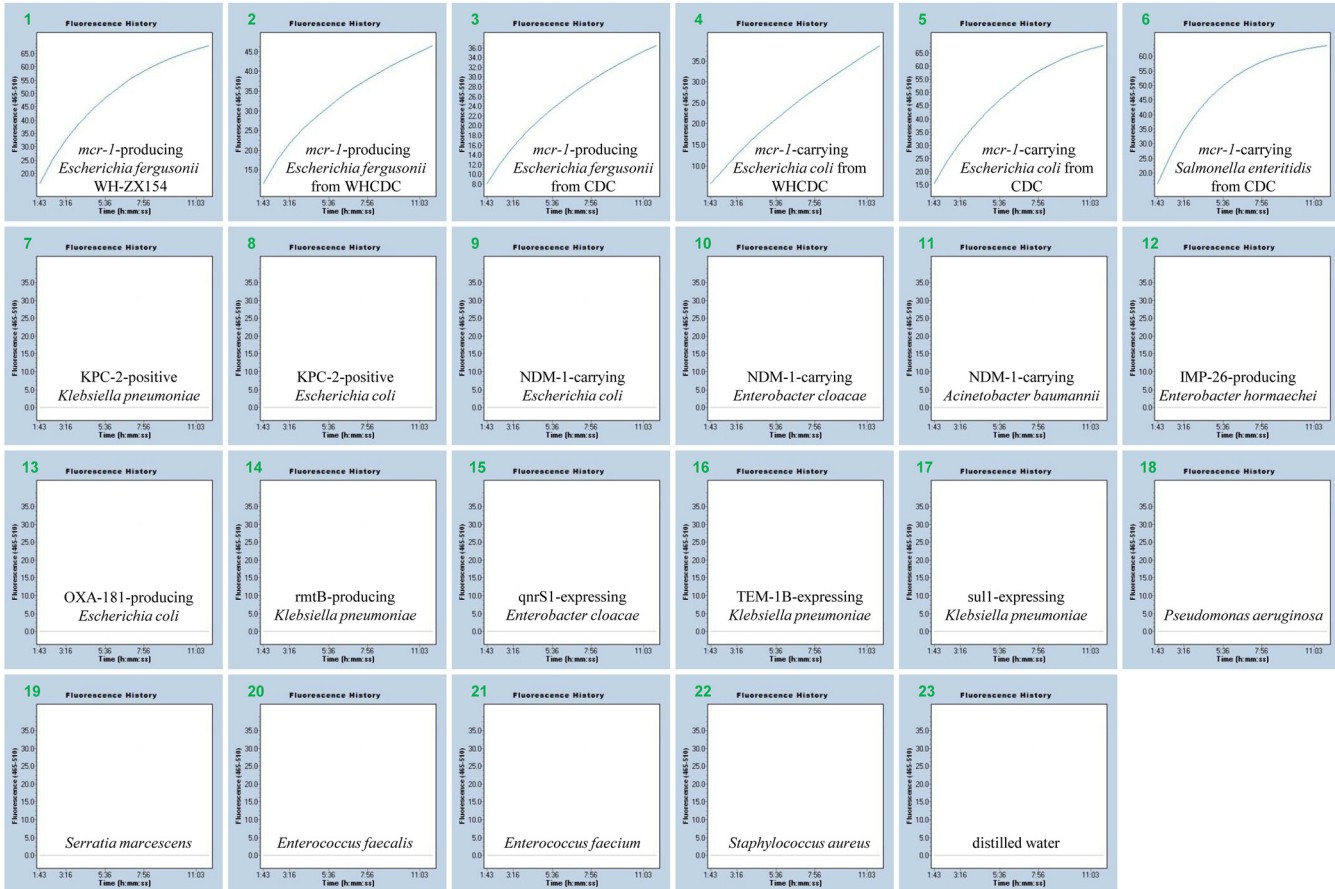

**FIG 6** Specificity of the *mcr-1* RPA-CRISPR/Cas12a assay. Graphs 1to 6, *Escherichia fergusonii* WH-ZX154, *Escherichia fergusonii* from WHCDC, *Escherichia fergusonii* from CDC, *Escherichia coli* from WHCDC, *Escherichia coli* from CDC, *Salmonella enteritidis* from CDC, separately (these isolates produced the *mcr-1* enzyme). Graphs 7 to 17, KPC-2-positive *Klebsiella pneumoniae*, KPC-2-positive *Escherichia coli*, NDM-1-carrying *Escherichia coli*, NDM-1-carrying *Enterobacter cloacae*, NDM-1-carrying *Acinetobacter baumannii*, IMP-26-producing *Enterobacter hormaechei*, OXA-181-producing *Escherichia coli*, rmtB-producing *Klebsiella pneumoniae*, qnrS1-expressing *Enterobacter cloacae*, TEM-1B-expressing *Klebsiella pneumoniae*, sul1-expressing *Klebsiella pneumoniae*, respectively. Graphs 18 to 22, *Pseudomonas aeruginosa*, *Serratia marcescens*, *Enterococcus faecalis*, *Enterococcus faecium*, *Staphylococcus aureus*, separately (these strains didn't carry aforementioned resistance genes). Graph 23 is distilled water.

## DISCUSSION

*Mcr-1* gene mediated by plasmid and transposon has widely disseminated in ecological environment (6), limiting the use of colistin in clinic and animal husbandry. In this study, we developed a simple and efficient identification assay coupling RPA with CRISPR/Cas12a system for the celerity diagnosis of *mcr-1* gene. In comparison with RPA, other isothermal amplification technologies such as MCDA and LAMP need intricated design of primers and are hard to augment multiple genes simultaneously (25). Therefore, we utilized RPA as a excellent means of nucleic acid amplification in our CRISPR/Cas12a assay.

A pair of specific primers cooperating with a gRNA sequence were employed to specially recognize the *mcr-1* gene, which could insure the high specificity of RPA-CRISPR/Cas12a assay. The specificity was notarized by means of identifying DNA extracted from *mcr-1*-positive species and negative strains. The intense fluorescence signals were merely observed in tubes loaded with target gene, but not in others. The PPV, NPV, sensitivity, and specificity of this method were all 100%. Hence, the diagnostic tool based on RPA-CRISPR/Cas12a for *mcr-1* detection showed a high accuracy. Besides specificity, the novel established assay appeared sensitive. The *mcr-1* RPA-CRISPR/Cas12a method could detect as little as 420 fg/μL of plasmid DNA in pure strains and approximately $1.6 \sim 6.2$ CFU/reaction ($1.6 \times 10^3 \sim 6.2 \times 10^3$ CFU/mL) in clinical specimens. The concentrations of genomes exacted from pure isolates are usually $1 \sim 200$ ng/μL, and the analytical sensitivity of our assay is less than 1 pg. Thus, the method will show positive reaction if the axenic culture stains carrying *mcr-1* gene.

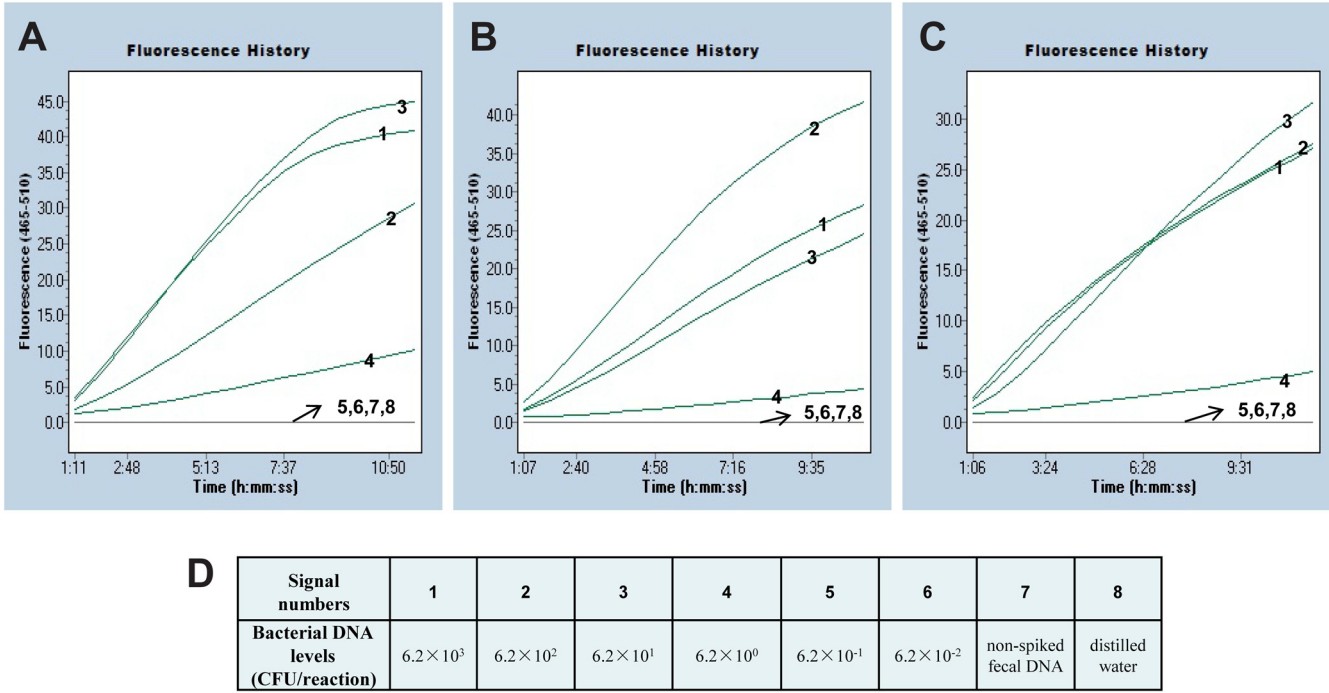

FIG 7 Detection limit of the RPA-CRISPR/Cas12a system in *mcr-1*-spiked Feces specimens. The information of three measurements was separately showed in Figures (A), (B) and (C). (D) Showed the signal number matching each bacterial concentration and control. Signals 1 to 8 represented the *Escherichia fergusonii* (WH-ZX154) DNA levels of $6.2 \times 10^3$, $6.2 \times 10^2$, $6.2 \times 10^1$, $6.2 \times 10^0$, $6.2 \times 10^{-1}$, and $6.2 \times 10^{-2}$ CFU per reaction, non-spiked fecal DNA, and distilled water, respectively.

The LoD of the built assay is 10 times as sensitive as that of conventional PCR (5). Although conventional PCR is the most frequently used method for *mcr-1* diagnosis, the insensitivity and time-consuming defects make it not suit for on-site detection. Relevant testing of LoD of *mcr-1* in actual samples is not found from document database, but the threshold in spiked clinical samples obtained by RPA-CRISPR/Cas12a assay has the same order of magnitude as that of previous studies (26, 27). In recent years, some methods based on isothermal amplification principle including LAMP, LAMP-LFB (Loop-mediated Isothermal Amplification Coupled with Lateral Flow Biosensor) and MCDA-LFB were developed for monitoring *mcr-1* (5, 28, 29), and the sensitivity of aforementioned assays described in previous researches are equal to that of the method established in this study. Noteworthily, the RPA-CRISPR/Cas12a assay is simpler in the matter of primers design and screening, and the fluorescence signal surveyed for readout is more objective and exact than LBF, which obtains the result through observing the test line and control line (30). The 2 red lines are vulnerable to aerosol contamination when the reaction tube is opened for soaking up amplicons, and then the false-positive result will be generated. In addition, we could find the unspecific fragments in Fig. 4, that's because nonspecific amplification is the inherent feature of RPA method (31, 32), and the accuracy of the test will not be significantly disturbed. Moreover, in the RPA-CRISPR/Cas12a system, gRNA can specifically recognize the target sequence of RPA products, so the experimental result is more reliable.

The entire experimental process of RPA-CRISPR/Cas12a method identifying *mcr-1* gene included rapid DNA extraction (15 min), RPA reaction (30 min), CRISPR/Cas12a cleavage (5 min), and fluorescence testing (10 min), which could be completed within 60 min. Therefore, the RPA-CRISPR/Cas12a assay saves detection time and is regarded as a useful tool for point-of-care testing.

In conclusion, a reliable RPA-CRISPR/Cas12a assay was devised for identification of *mcr-1* gene causing colistin resistance, which offers an excellent guide for anti-infection therapy. The RPA-CRISPR/Cas12a approach established here was rapid,

**TABLE 1** Strains used in this study

| Genotype[a] | Bacteria species | Source of strains[b] | No. of isolates |
|---|---|---|---|
| *mcr-1* | *Escherichia fergusonii* | WHCDC (WH-ZX154) | 1 |
| | *Escherichia fergusonii* | WHCDC | 2 |
| | *Escherichia fergusonii* | CDC | 2 |
| | *Escherichia coli* | WHCDC | 6 |
| | *Escherichia coli* | CDC | 3 |
| | *Salmonella enteritidis* | CDC | 1 |
| KPC-2 | *Klebsiella pneumoniae* | WHCDC | 8 |
| | *Escherichia coli* | WHCDC | 3 |
| NDM-1 | *Escherichia coli* | WHCDC | 4 |
| | *Enterobacter cloacae* | WHCDC | 2 |
| | *Acinetobacter baumannii* | WHCDC | 3 |
| IMP-26 | *Enterobacter hormaechei* | WHCDC | 2 |
| OXA-181 | *Escherichia coli* | WHCDC | 1 |
| rmtB | *Klebsiella pneumoniae* | WHCDC | 8 |
| qnrS1 | *Enterobacter cloacae* | WHCDC | 2 |
| TEM-1B | *Klebsiella pneumoniae* | WHCDC | 5 |
| sul1 | *Klebsiella pneumoniae* | WHCDC | 4 |
| Non | *Pseudomonas aeruginosa* | WHCDC | 4 |
| | *Serratia marcescens* | WHCDC | 2 |
| | *Enterococcus faecalis* | WHCDC | 3 |
| | *Enterococcus faecium* | WHCDC | 1 |
| | *Staphylococcus aureus* | WHCDC | 3 |

[a]Non, the strains didn't carry aforementioned resistance genes.
[b]CDC, Chinese Center for Disease Control and Prevention. WHCDC, Wuhan Center for Disease Control and Prevention.

specific, and highly sensitive. The whole process was accomplished within 1 h, and this assay could detect 420 fg plasmid DNA per reaction in pure culture. Moreover, it is able to accurately divide *mcr-1* and non-*mcr-1* samples. Thus, the RPA-CRISPR/Cas12a assay built in this study provides a rapid diagnosis way for monitoring *mcr-1* gene in clinic and livestock farm.

## MATERIALS AND METHODS

**Bacterial strains.** Fifteen *mcr-1*-positive and 55 non-*mcr-1* isolates were collected in this study (Table 1). The *mcr-1*-producing bacterial organisms included *S. enteritidis*, *E. coli* and *E. fergusonii*, 8 kinds of resistance genes consisted of KPC-2, NDM-1, IMP-26, OXA-181, rmtB, qnrS1, TEM-1B, and sul1 were incorporated in the *mcr-1*-negative strains. The first 4 genes mediated carbapenem resistance, and the action objects of the rest were aminoglycosides, quinotones, extended-spectrum beta-lactams, and tetracyclines, respectively. All genes were verified using PCR and sequencing. On the basis of operating instruction, genome, plasmid, and clinical samples DNA were separately extracted by the corresponding kit (Qiagen Co., Ltd.). The plasmid DNA of *E. fergusonii* (WH-ZX154) carrying *mcr-1* gene was regarded as template to optimize the RPA reaction conditions and explore the threshold of CRISPR/Cas12a assay. The RPA amplification was executed using recombinase-mediated isothermal amplification kit, which was composed of reaction buffer, enzyme mixture and reaction trigger (HuiDeXin Biotechnology Development Co., Ltd.). The isothermal condition of RPA reaction was supplied by a heater (MTH-100, MiU Instruments Co., Ltd.). Afterwards the RPA products were added to CRISPR/Cas12a reaction system (Magigen Biotechnology Co., Ltd.), and the obtained reaction mixtures emitting fluorescence could be detected in real-time PCR(Roche LightCycler 480, Roche).

**Oligonucleotides and gRNA preparation.** A pair of primers for RPA reaction targeted *mcr-1* gene (GenBank KX458104.1) were designed by Primer Premier 6.0 software to match the CRISPR/Cas12a detection. The forward primer 5′-GATGATTTCATCGCTCAAAGTATCCAGTGGC-3′ and reverse primer 5′-

```
                                             F
GCCTTGCTTGCCACCGATGATTTCATCGCTCAAAGTATCCAG

TGGCTGCAGACGCACAGCAATGCCTATGATGTCTCAATGCT
           ──────────────────▶
                   gRNA
GTATGTCAGCGATCATGGCGAAAGTCTGGGTGAGAACGGTG
                               ◀──────────────────
TCTATCTACATGGTATGCCAAATGCCTTTGCACCAAA
     R
```

**FIG 8** Location of the primers and gRNA on the partial *mcr-1* gene segment. The arrowed lines represent the amplification orientation of primers.

**TABLE 2** gRNA, probe, and primers of the RPA-CRISPR/Cas12a assay to detect the *mcr-1* gene

| Objects | Sequences and modifications (5′–3′) | Length[b] |
|---|---|---|
| gRNA | UAAUUUCUACUAAGUGUAGAUCAGACGCACAGCAAUGCCUAGG | 41 mer |
| Probe[a] | FAM-TATTAT-BHQ1 | 6 mer |
| Primer-F | GATGATTTCATCGCTCAAAGTATCCAGTGGC | 31 nt |
| Primer-R | CATACCATGTAGATAGACACCGTTCTCACC | 30 nt |

[a]5′ end of probe was labeled with FAM, and 3′ end was labeled with BHQ1.
[b]nt referred to nucleotide, mer meant monomeric unit.

CATACCATGTAGATAGACACCGTTCTCACC-3′ were analyzed using BLAST tool for specificity verification. Moreover, the gRNA and probe were designed according to the *mcr-1* RPA-CRISPR/Cas12a principle. The RPA primers and gRNA were designed in the conservative DNA region of the *mcr-1* gene (Fig. S4) and whose positions were shown in Fig. 8, and the three sequences described above were displayed in Table 2. Furthermore, the probe was labeled with FAM fluorophore at 5′end and BHQ1 quencher at 3′end. All oligo fragments were synthesized and purified by Sangon Biotechnology Co., Ltd.

**RPA amplification.** In accordance with the operation manual, the amplification of the *mcr-1* gene was carried out using a commercial RPA kit. In brief, a 25 $\mu$L reaction system including buffer I (13 $\mu$L), buffer II (1.25 $\mu$L), enzyme mixture (2.5 $\mu$L), primer (0.4 $\mu$M each), reaction trigger (1.25 $\mu$L), DNA template (1 $\mu$L), and distilled water (3.6 $\mu$L) was incubated at 37°C for 30 min. IMP-26-positive *E. hormaechei* (WHCDC-SP67), qnrS1-producing *E. cloacae* (WHCDC-XW38) and distilled water were served as references. To obtain a better reaction temperature, the RPA reaction was executed in the range of 35 to 42°C with 1°C intervals.

**CRISPR/Cas12a detection.** gRNA was mixed with Cas12a enzyme to make CRISPR/Cas12a-gRNA complex for trimming the RPA product. The CRISPR/Cas12a cleavage reaction was implemented in a 50 $\mu$L mix consisting of 5 $\mu$L 10xreaction buffer, 0.4 $\mu$L gRNA (10 $\mu$M), 0.4 $\mu$L ssDNA probe (10 $\mu$M), 0.2 $\mu$L Cas12a protein (20 $\mu$M), 2 $\mu$L RPA product, and 42 $\mu$L of nuclease-free water. The reaction process needed to take 5 min at 37°C. Real-time fluorescence detector was used to analyze the fluorescent signal in the end.

**Sensitivity and specificity evaluation of the RPA-CRISPR/Cas12a method.** To evaluate the LoD of the RPA-CRISPR/Cas12a assay with *mcr-1*, the plasmid DNA of *E. fergusonii* WH-ZX154 was serially diluted from 4.2 ng/$\mu$L to 4.2 fg/$\mu$L, negative control and blank control were performed simultaneously. Real-time fluorescence was analyzed to obtain the result. The assay was confirmed 3 times to validate stability. The specificity of RPA-CRISPR/Cas12a assay was evaluated with *mcr-1*-positive strains and non-*mcr-1* bacteria (Table 1). Each test was repeated twice.

**RPA-CRISPR/Cas12a detection in spiked clinical specimens.** The feasibility of RPA-CRISPR/Cas12a system was assessed using spiked clinical specimens including stool, blood, and urine samples. For spiked stool samples, operational process was performed as previously described (29), the reference isolate *E. fergusonii* WH-ZX154 was 10-fold serially diluted ($6.2 \times 10^6 \sim 6.2 \times 10^1$ CFU/mL), 100 $\mu$L of each dilution was added into individual fecal specimen (0.2g). Similarly, the blood sample (200 $\mu$L) was added with 100 $\mu$L of reference isolate ($2.6 \times 10^6 \sim 2.6 \times 10^1$ CFU/mL), respectively. And the urine sample (900 $\mu$L) was severally added with strain ($1.6 \times 10^6 \sim 1.6 \times 10^1$ CFU/mL). The stool, blood and urine DNA of spiked samples were extracted and dissolved in 100 $\mu$L of elution buffer, and 1 $\mu$L of solution was used for RPA-CRISPR/Cas12a detection as templates. Thus, the contents of stool DNA using for reactions were $6.2 \times 10^3 \sim 6.2 \times 10^{-2}$ CFU/reaction according to the conversion. Analogously, blood DNA were $2.6 \times 10^3 \sim 2.6 \times 10^{-2}$ CFU/reaction, and urine DNA were $1.6 \times 10^3 \sim 1.6 \times 10^{-2}$ CFU/reaction. The non-spiked specimen DNA and distilled water were served as control. Finally, 3 measurements were used to determine the LoD of RPA-CRISPR/Cas12a system in simulation samples.

**Ethics statement.** The study was approved by the Ethics Committee of Wuhan Center for Disease Control and Prevention (WHCDCIRB-K-2021038). The clinical specimens were collected from a healthy man who had signed the informed consent. All experimental processes were operated in BSL-2 lab.

## SUPPLEMENTAL MATERIAL

Supplemental material is available online only.
**SUPPLEMENTAL FILE 1**, PDF file, 0.8 MB.

## ACKNOWLEDGMENTS

This study was funded by the Beijing Nova Program (Z211100002121042), Project of Medical Research of Wuhan (WG17Q02 and WG21Z07), and Natural Science Foundation of Hubei Province (2019CFC899).

We thank Jie Che and Juan Li for providing specimens.

L.G. and Y.W. designed the assay. J.L. and F.T. directed the orientation of research. L.G., F.Y., Z.J. and E.L. executed the experiments. Z.J. provided partial samples. Y.W. contributed the reagents. L.G., E.L. and X.L. analyzed the data. L.G. and X.L. wrote the manuscript. Y.W. reviewed the manuscript.

The authors denote no conflict of interest.

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
