## [Reviewer comments · Microbiology Spectrum]

Microbiology Spectrum

Highly Sensitive and Specific Detection of Mobilized Colistin Resistance Gene *mcr-1* by CRISPR-Based Platform

Lin Gong, Zhengjiang Jin, Ernan Liu, Fei Tang, Fengyun Yuan, Jiansheng Liang, Yimei Wang, Xiaoli Liu, and Yi Wang

Corresponding Author(s): Yi Wang, Capital Institute of Pediatrics

Review Timeline:

Submission Date:	May 20, 2022
Editorial Decision:	June 27, 2022
Revision Received:	August 1, 2022
Accepted:	August 15, 2022

Editor: Florence Doucet-Populaire

Reviewer(s): The reviewers have opted to remain anonymous.

Transaction Report:

DOI: <https://doi.org/10.1128/spectrum.01884-22>

June 27, 2022

Dr. Yi Wang
Capital Institute of Pediatrics
Beijing
China

Re: Spectrum01884-22 (Highly Sensitive and Specific Detection of Mobilized Colistin Resistance Gene *mcr-1* by CRISPR-Based Platform)

Dear Dr. Yi Wang:

Link Not Available

Sincerely,

Florence Doucet-Populaire

Journals Department
Reviewer comments:

Reviewer #1 (Comments for the Author):

The authors present a significant work that provides a new method for on-site detection of *mcr-1* gene. The method integrates RPA and CRISPR/Cas12a as rapid molecular testing, which validation was made from different types of samples (bacterial DNA and fecal samples). The manuscript is well written, and the authors use a good experimental design to support their conclusions. However, I suggest addressing the following corrections for clarity.

Keep the term "gRNA" in lowercase since it is sometimes incorrectly written as "GRNA".

Methods: agarose gel electrophoresis should be explained. At least the agarose % and amount of sample loaded on the gel.

Fig5. I suggest labeling the lanes with the respective temperature per lane. Regarding this result, the authors should discuss the unspecific fragments observed in the RPA product and how they might interfere or not in the detection method.

Fig6, 7, and 8 Legends. It is difficult to follow the number that matches the treatment. I suggest being more specific in the legend and adding the number matching each plasmid concentration, blank and negative controls. The authors have enough space on the graphics of fig7 to add the name of the treatment instead of numbers. In Fig 8, do not refer to Fig6 legend. Each legend should be clear enough to understand the figure.

For Mcr-1-spiked Feces Samples, the author explained the methods and results in terms of CFU/ml preparation, but Fig8 show them in CFU per reaction, then it is hard to understand the description looking at the figure. Also, in the legend, by transforming the units to CFU/reaction, the authors test for negative concentrations (6.2×10^{-1} , and 6.2×10^{-2}). How could you expect a detection in concentration below 1 CFU? At least 1 CFU will be needed to provide DNA. The concentrations between 1-6.2 CFU/reaction would have been more interesting to establish a detection limit.

Lines 251-269: This was already said in the introduction and is not completing a discussion, I suggest avoiding redundancy and going straight to the discussion.

245: discuss if this amount can be found on actual samples compared with other studies.

239, 277: avoid the subjective adjectives: "great sensitivity", "ultrasensitive" , without a parameter to classify the method as mentioned.

Fig6 results: The authors should discuss how they relate this result with the number of bacteria required for the detection in a real the sample.

The discussion could improve, as it is highly redundant in the summary of results. In addition, a comparison with other methods, sample types, and the current usage of this method for detecting other microbes might be helpful for readers.

Review minor English mistakes, for example "nuclic acid" in line 91; phrase "Analysing 171 real-time fluorescence could obtain the result." In line 171.

Reviewer #2 (Comments for the Author):

Spectrum01884-22

The manuscript title "Highly Sensitive and Specific Detection of Mobilized Colistin Resistance Gene mcr-1 by CRISPR-Based Platform" is interesting and innovative. The manuscript is well written. CRISPR Cas technology applied to the diagnosis of the infectious diseases show an important perspective. I have some comments to take in consideration to the authors:

Major comments

- Add the negative and positive predictive values (PPV and NPV) of the RPA-CRISPR/Cas12a technology to mcr-1 detection.
- Which is the sensitivity and specific values of the RPA-CRISPR/Cas12a technology? The authors only analyze the limit of the detection.
- The authors described very well the technique but not develop of the analyze in the different clinical samples. It would be necessary to carry out clinical samples artificially contaminated.
- RPA primers and gRNA were designed in the conservative DNA region of the mcr-1 gene from different pathogens? Did you make an alignment the sequences? Please, clarify it.

Staff Comments:

Preparing Revision Guidelines

Please return the manuscript within 60 days; if you cannot complete the modification within this time period, please contact me. If you do not wish to modify the manuscript and prefer to submit it to another journal, please notify me of your decision immediately so that the manuscript may be formally withdrawn from consideration by Microbiology Spectrum.

The authors present a significant work that provides a new method for on-site detection of *mcr-1* gene. The method integrates RPA and CRISPR/Cas12a as rapid molecular testing, which validation was made from different types of samples (bacterial DNA and fecal samples). The manuscript is well written, and the authors use a good experimental design to support their conclusions. However, I suggest addressing the following corrections for clarity.

Keep the term “gRNA” in lowercase since it is sometimes incorrectly written as “GRNA”.

Methods: agarose gel electrophoresis should be explained. At least the agarose % and amount of sample loaded on the gel.

Fig5. I suggest labeling the lanes with the respective temperature per lane. Regarding this result, the authors should discuss the unspecific fragments observed in the RPA product and how they might interfere or not in the detection method.

Fig6, 7, and 8 Legends. It is difficult to follow the number that matches the treatment. I suggest being more specific in the legend and adding the number matching each plasmid concentration, blank and negative controls. The authors have enough space on the graphics of fig7 to add the name of the treatment instead of numbers. In Fig 8, do not refer to Fig6 legend. Each legend should be clear enough to understand the figure.

For *Mcr-1*-spiked Feces Samples, the author explained the methods and results in terms of CFU/ml preparation, but Fig8 show them in CFU per reaction, then it is hard to understand the description looking at the figure. Also, in the legend, by transforming the units to CFU/reaction, the authors test for negative concentrations (6.2×10^{-1} , and 6.2×10^{-2}). How could you expect a detection in concentration below 1 CFU? At least 1 CFU will be needed to provide DNA. The concentrations between 1-6.2 CFU/reaction would have been more interesting to establish a detection limit.

Lines 251-269: This was already said in the introduction and is not completing a discussion, I suggest avoiding redundancy and going straight to the discussion.

245: discuss if this amount can be found on actual samples compared with other studies.

239, 277: avoid the subjective adjectives: “great sensitivity”, “ultrasensitive”, without a parameter to classify the method as mentioned.

Fig6 results: The authors should discuss how they relate this result with the number of bacteria required for the detection in a real the sample.

The discussion could improve, as it is highly redundant in the summary of results. In addition, a comparison with other methods, sample types, and the current usage of this method for detecting other microbes might be helpful for readers.

Review minor English mistakes, for example “nuclic acid” in line 91; phrase “Analysing 171 real-time fluorescence could obtain the result.” In line 171.

Dear Reviewers,

Thank you for your letter and for the comments concerning our manuscript entitled “**Highly Sensitive and Specific Detection of Mobilized Colistin Resistance Gene *mcr-1* by CRISPR-Based Platform**” (ID: Spectrum01884-22). Those comments are valuable and very helpful for revising and improving our paper, as well as the important guiding significance to our researches. We have studied comments carefully and have made correction which we hope meet with approval. Revised portion are marked in the paper. The main corrections in the paper and the responds to the reviewers’ comments are as flowing:

Reviewer #1:

1. Keep the term "gRNA" in lowercase since it is sometimes incorrectly written as "GRNA".

Responds: We have made correction according to the reviewer’s comment.

2. Methods: agarose gel electrophoresis should be explained. At least the agarose % and amount of sample loaded on the gel.

Responds: The relevant details have been added into the manuscript.

3. Fig. 5. I suggest labeling the lanes with the respective temperature per lane. Regarding this result, the authors should discuss the unspecific fragments observed in the RPA product and how they might interfere or not in the detection method.

Responds: Each lane has been labeled with the respective temperature in Figure 5. We could find the unspecific fragments in Figure 5, that's because nonspecific amplification is the inherent feature of RPA method (Piepenburg et al., 2006; Wang et al., 2020), and the accuracy of the test will not be significantly disturbed. Moreover, in the RPA-CRISPR/Cas12a system, gRNA can specifically recognize the target sequence of RPA products, so the experimental result is more reliable. The relevant details have been discussed in the manuscript.

4. Fig. 6, 7, and 8 Legends. It is difficult to follow the number that matches the treatment. I suggest being more specific in the legend and adding the number matching each plasmid concentration, blank and negative controls. The authors have

enough space on the graphics of Fig. 7 to add the name of the treatment instead of numbers. In Fig 8, do not refer to Fig6 legend. Each legend should be clear enough to understand the figure.

Responds: We have made correction according to the reviewer's comment.

5. For *mcr-1*-spiked feces samples, the author explained the methods and results in terms of CFU/ml preparation, but Fig. 8 show them in CFU per reaction, then it is hard to understand the description looking at the figure. Also, in the legend, by transforming the units to CFU/reaction, the authors test for negative concentrations (6.2×10^{-1} , and 6.2×10^{-2}). How could you expect a detection in concentration below 1 CFU? At least 1 CFU will be needed to provide DNA. The concentrations between 1-6.2 CFU/reaction would have been more interesting to establish a detection limit.

Responds: Actually, 6.2×10^{-1} and 6.2×10^{-2} CFU/reaction were respectively corresponded to 6.2×10^2 and 6.2×10^1 CFU/mL of reference isolates. 100 μ L of each dilution was added into individual fecal specimen (0.2g), so the stool DNA of spiked samples were extracted from 62 and 6.2 CFU isolates correspondingly. The extracted genomic DNA was dissolved in 100 μ l of elution buffer, and 1 μ l of which was used for RPA-CRISPR/Cas12a detection as templates. Thus, the contents of DNA using for reaction were respectively 6.2×10^{-1} and 6.2×10^{-2} CFU according to the conversion.

Generally, the detection limit is accurate to the order of magnitude, and what the significant number is doesn't mean much. Moreover, other clinical samples including blood and urine specimens have been spiked with the reference isolates, which could obtain various detection limits to verify the feasibility of RPA-CRISPR/Cas12a system.

6. Lines 251-269: This was already said in the introduction and is not completing a discussion, I suggest avoiding redundancy and going straight to the discussion.

Responds: The section has been streamlined.

7. 245: discuss if this amount can be found on actual samples compared with other studies.

Responds: Relevant testing of LoD of *mcr-1* in actual samples is not found from document database, but the threshold in spiked clinical samples obtained by RPA-CRISPR/Cas12a assay has the same order of magnitude as that of previous studies (Bontron et al., 2016; Wang et al., 2017). The relevant details have been discussed in the manuscript.

8. 239, 277: avoid the subjective adjectives: "great sensitivity", "ultrasensitive" , without a parameter to classify the method as mentioned.

Responds: We have made correction according to the reviewer's comment.

9. Fig. 6 results: The authors should discuss how they relate this result with the number of bacteria required for the detection in a real the sample.

Responds: The concentrations of genomes exacted from pure stains are usually 1 ~ 200 ng/μL, and the analytical sensitivity of our assay is 420 fg/μL in pure *mcr-1*-positive isolates. Thus, The method will show positive reaction if the axenic culture stains carrying *mcr-1* gene. The relevant details have been discussed in the manuscript.

10. The discussion could improve, as it is highly redundant in the summary of results. In addition, a comparison with other methods, sample types, and the current usage of this method for detecting other microbes might be helpful for readers.

Responds: Considering the reviewer's suggestions, the blood and urine samples have been spiked to analyse the sensibility of the assay. Some methods based on isothermal amplification principle including LAMP, LAMP-LFB (Loop-mediated Isothermal Amplification Coupled with Lateral Flow Biosensor) and MCDA-LFB were developed for monitoring *mcr-1*, these methods have been compared with RPA-CRISPR/Cas12a system in this manuscript.

11. Review minor English mistakes, for example "nuclie acid" in line 91; phrase "Analysing real-time fluorescence could obtain the result." In line 171.

Responds: We have made correction according to the reviewer's comment.

Reviewer #2:

1. Add the negative and positive predictive values (PPV and NPV) of the RPA-CRISPR/Cas12a technology to *mcr-1* detection.

Responds: PPV and NPV have been added (TABLE S1).

2. Which is the sensitivity and specific values of the RPA-CRISPR/Cas12a technology? The authors only analyze the limit of the detection.

Responds: Sensitivity and specific values have been added (TABLE S1).

3. The authors described very well the technique but not develop of the analyze in the different clinical samples. It would be necessary to carry out clinical samples artificially contaminated.

Responds: Other clinical samples (blood and urine samples) have been spiked to analyse the sensibility of the assay.

4. RPA primers and gRNA were designed in the conservative DNA region of the *mcr-1* gene from different pathogens? Did you make an alignment the sequences? Please, clarify it.

Responds: RPA primers and gRNA were designed in the conservative DNA region of the *mcr-1* gene. The sequences of *mcr-1* gene from different isolates were aligned in Figure S1.

REFERENCE

- Piepenburg, O., Williams, C. H., Stemple, D. L., and Armes N. A. (2006). DNA detection using recombination proteins. *PLoS Biol.* 4(7): e204. doi: 10.1371/journal.pbio. 0040204.
- Wang, Y., Jiao, W., Wang, Yu., Wang, Ya., Shen, C., Qi, H., et al.(2020). Graphene oxide and self-avoiding molecular recognition systems-assisted recombinase polymerase amplification coupled with lateral flow bioassay for nucleic acid detection. *Microchim. Acta* 187:667. doi: 10.1007/s00604-020-04637-5.
- Bontron, S., Poirel, L., and Nordmann, P. (2016). Real-time PCR for detection of plasmid-mediated polymyxin resistance (*mcr-1*) from cultured bacteria and

stools. *J. Antimicrob. Chemother.* 71, 2318-2320. doi: 10.1093/jac/dkw139.

Wang, Y., Li, H., Wang, Y., Zhang, L., Zhang, J., Xu, J., et al. (2017). Nanoparticle-based lateral flow biosensor combined with multiple cross displacement amplification for rapid, visual and sensitive detection of *Vibrio cholerae*. *FEMS Microbiol. Lett.* 364, fnx234. doi::10.1093/femsle/fnx234.

We tried our best to improve the manuscript and made some changes in the manuscript. And we appreciate for the reviewers' warm work earnestly, and hope that the correction will meet with approval.

Once again, thank you very much for your comments and suggestions.

Sincerely,

Lin Gong

Yi Wang

August 12, 2022

Dr. Yi Wang
Capital Institute of Pediatrics
Beijing
China

Re: Spectrum01884-22R1 (Highly Sensitive and Specific Detection of Mobilized Colistin Resistance Gene *mcr-1* by CRISPR-Based Platform)

Dear Dr. Yi Wang:

Your manuscript has been accepted, and I am forwarding it to the ASM Journals Department for publication. You will be notified when your proofs are ready to be viewed.

Sincerely,

Florence Doucet-Populaire
Editor, Microbiology Spectrum
